# Dynamic Scenario Predictions of Peak Carbon Emissions in China's Construction Industry

**Xilian Wang [1], Lihang Qu [1,]*, Yueying Wang [2] and Helin Xie [3]**

1    Energy Economics and Management Research Centre, College of Management, Xi'an University of Science and Technology, Xi'an 710054, China; wxllian912428@126.com
2    School of Foreign Languages, Northwest University, Xi'an 710127, China
3    School of Civil Engineering, Sun Yat-sen University, Zhuhai 519082, China
*    Correspondence: qlhang0422@163.com

**Abstract:** As the largest carbon emitter in the world, China aims to reach its peak carbon emissions goal by the year 2030, while the construction industry makes a significant contribution to carbon emissions, directly affecting the country's commitment to meet its target. The present paper investigates the dynamic characteristics of carbon emissions released by China's construction industry under single- and multiple-scenario settings with altering economic growth rates, optimizing energy structures, adjusting industrial structures, and modifying carbon emission policy factors. The research results show that the total carbon emissions generally present a steady increase from the year 2000 and will reach 12,880.40 million tons (MT) by 2030 under a scenario without any intervention. Indirect carbon emissions released from associated industries account for over 96% of the total carbon emissions, while direct carbon emissions make a minor contribution to the total. Single and comprehensive scenarios have positive effects on reducing emissions; it was also observed that only under energy structure scenario III and comprehensive scenario III could carbon emissions released from the construction sector reach a peak value by 2030. The effects of emissions reductions as a result of single policies can be presented in the following order: energy structure, economic growth, carbon emissions policy factor, and industrial structure. All of the emissions reduction effects of multiple scenarios are superior to the single scenarios. The research results provide a basis and guidance for policymakers to adopt the correct steps to fulfill China's aim of achieving peak carbon emissions by the projected date.

**Keywords:** carbon emissions; construction industry; system dynamics model; scenario simulation

## 1. Introduction

The influence of global warming and climate change on the environment and the survival of human beings has become increasingly worse in recent years. Carbon emissions are the main cause of these problems; therefore, the reduction of these carbon emissions has become an urgent topic in the attempt to solve the global warming crisis. China has been responsible for one-third of the world's carbon emissions and has also played an essential role in the advancement of mitigation efforts. At the 75th session of the United Nations General Assembly in September 2020, Xi Jinping made a solemn promise that China would endeavor to reach a peak in its total carbon emissions by 2030 and attain carbon neutrality by the year 2060. The 14th Five-Year Plan of China, which was released in March 2021, stated that China would support regions and key industries in taking the initiative to reach an emissions peak and stimulate a low-carbon transition in the fields of industry, construction, and transportation.

The construction industry accounts for approximately 40% of global energy consumption and 36% of global greenhouse gas (GHG) emissions [1], while it utilizes over 20% of energy consumption and releases 25% of GHG emissions in China [2]. According to

the China Building Energy Consumption Research Report (2021), the $CO_2$ emissions produced by the entire construction process were 4.997 billion tons (BT), making up 50.6% of the national total carbon emissions. A total of 0.1 and 2.13 BT of waste were released during the building and operation phases, respectively, which accounted for 22.6% of all emissions released (see Figure 1). The assessment report conducted by the Fourth Intergovernmental Panel on Climate Change projected that the energy consumption of new and old buildings may be lowered by 30–50% without considerably increasing investment costs [3]. As a result, a key point in the evolution of the construction industry is to reduce carbon emissions.

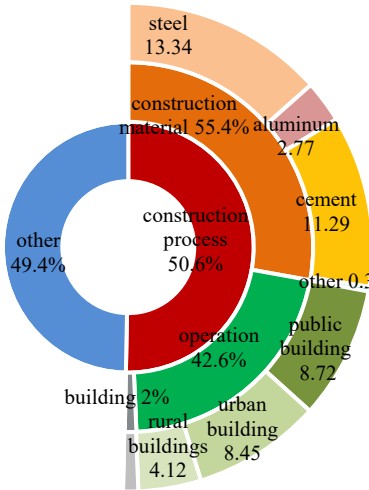

**Figure 1.** The proportion of carbon emissions produced from the entire construction process in 2019. The unit is a hundred million tons of $CO_2$.

## 2. Literature Review

Some scholars in the field have focused their research on primarily identifying influential factors and subsequently developed a prediction model to provide projections for energy consumption and carbon emissions. These prediction models include the Stochastic Impacts by Regression on Population, Affluence and Technology (STIRPAT) model, the factor decomposition model, and other models. The Impact, Population, Affluence, and Technology (IPAT) model was primarily used in the research to depict the various impacts of the population, affluence, and technology on the environment [4]. For example, Lai et al. (2019) used a modified Kaya model in their research to examine the relationship and trends evident between carbon emissions, energy consumption, GDP growth rate, and carbon intensity in China's construction industry [5]. Lin and Wang (2019) used Kaya's extended model to dynamically analyze the effects of investment scale, investment efficiency, energy intensity, energy structure, and carbon emission factors on carbon emissions in China's coke manufacturing industry [6]. The STIRPAT model, an extension of the IPAT model, may be extensively utilized to evaluate the intricate interactions between human social systems and the natural environment [7,8].

The factor decomposition analysis model is employed to inquire into the contribution of each factor to the change in a dependent variable [9]. Index decomposition analysis (IDA) and structural decomposition analysis (SDA) are two types of factor decomposition models [10]. Lin and Liu (2015) used the logarithmic mean Divisia index (LMDI) decomposition method in their work to study the economic factors of carbon emissions released from commercial and residential buildings in China. They observed that the change in energy-related carbon emissions produced by commercial buildings displays the characteristics of the environmental Kuznets curve [11]. Changes in key performance indicators, such as energy consumption and carbon emissions, have been analyzed in the literature using the SDA [12]. The generalized divisor index method (GDIM) and the computable

general equilibrium (CGE) model are two additional models that are used in the process of determining the impact variables that have an effect on carbon emissions [13,14].

The majority of the studies that have been performed to predict energy consumption and carbon emissions in the construction sector use scenario prediction methods and gray prediction models. The scenario prediction method analyzes the macro-environment and predicts the dependent variable by analyzing changes occurring in the model's independent variables under various scenarios. Representative models include the STIRPAT model, the linear regression model, and others. In their study, Ji and Jiang (2012) investigated the relationship between the average annual growth rate of energy consumption per unit of added value in China's construction industry and carbon emissions, emissions reduction effect, and peak time by using the STIRPAT model and a scenario analysis [15]. Wang (2013) used the Kaya formula to predict the carbon emissions of China's construction industry from 2011 to 2020 and provided policy recommendations [16]. The purpose of gray prediction is to assess whether the development trend among system elements can be successfully established. Then, a regular data series may be produced to anticipate the future trend of development by processing the original data to determine the pattern of system changes. Wu et al. (2015) used a novel multivariate gray model to simulate and predict carbon emissions in BRICS countries (Brazil, Russia, India, China, and South Africa) [17].

Additional methods have also been used to perform predictions in the construction sector. For example, Ma et al. (2019) used the long-term energy alternative planning system (LEAP) model to simulate and forecast historical spatio-temporal highway-passenger-transit carbon emissions in the Beijing–Tianjin–Hebei urban areas from 2005 to 2014, as well as future emissions up to the year 2030 under various scenarios [18]. Cheng et al. (2016) used a regional CGE model to analyze the impact of low-carbon policies in the power sector of Guangdong Province on its energy and carbon emissions targets by 2020 [19]. Li and Gao (2018) constructed the improved particle swarm optimization-back propagation (IPSO-BP) model based on 44 scenarios in the second generation of new dry-cement technology systems to anticipate the peak of carbon emissions produced by China's cement industry between 2016 and 2050 [20].

Although the scenario prediction analysis and the gray prediction model can be used to identify the peak value of carbon emissions, they are merely static predictions that do not consider the uncertainty of future changes in the variables. In contrast, the system dynamics (SD) method combines qualitative and quantitative analyses to characterize these undefined behaviors using systematic synthesis reasoning, which makes it the optimal choice for addressing complex, nonlinear system issues. Additionally, the SD model has been applied to various fields. At the industry level, it has been widely used to study system issues, such as transportation and agriculture [21–23]. At the regional level, it has also been applied to investigate national and urban energy consumption, carbon emissions, and potential policies [24,25]. Therefore, this study adopts a SD model to comprehensively investigate the characteristics of carbon emissions in the construction sector.

In summary, previous research has provided us with some relevant models and evidence for the impact factors and carbon emission predictions; however, to date, most models find it difficult to explain the interaction and dynamic feedback mechanisms occurring between the various levels of factors. Fewer studies concentrate on carbon emissions produced by the construction industry, especially regarding its peak value; both the simulations and scenario analyses of carbon emissions have not accurately reflected the impact on the uncertainty and randomness of different policy variables in the literature.

## 3. Methodology and Data

The present paper adopted the carbon emissions obtained from China's construction industry as the research object and set the coverage period of the model from 2005 to 2030 with a time step of one year. The SD model was used to analyze the carbon emissions evolution mechanism of the construction industry. The carbon emission factor and input–output methods were then used to forecast the carbon emissions produced by the construction

sector, and a scenario analysis method was used to simulate and predict the dynamic change characteristics of carbon emissions. Figure 2 depicts the research framework and technological path.

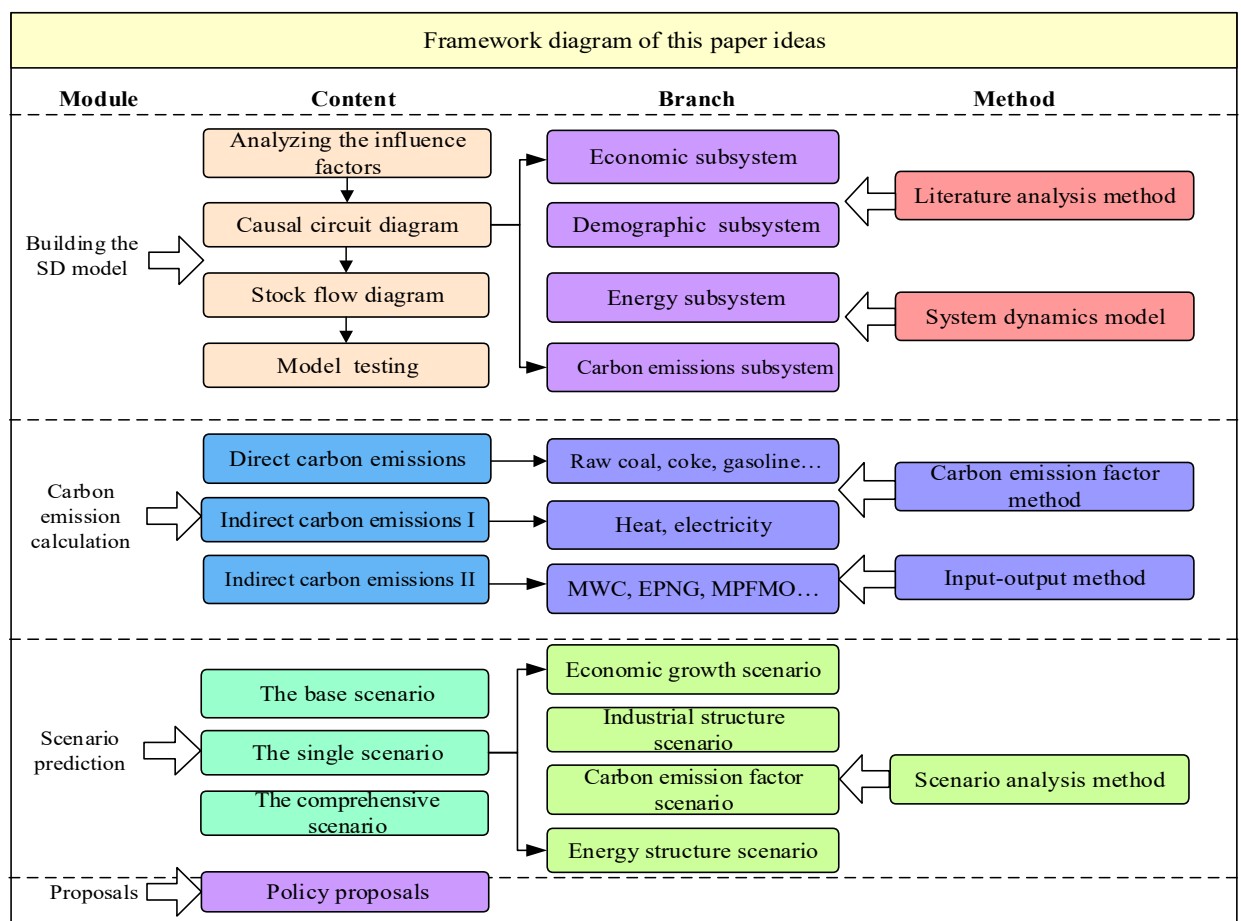

**Figure 2.** Framework diagram of ideas. Detailed meanings of MWC, EPNG, and MPFMO are presented in Section 3.2.

### 3.1. SD Model

The SD model was developed in 1956 as a means of coping with complex problems in social, economic, and ecological systems in a more effective manner [26]. Based on the feedback control theory, it extensively relies on computer simulation techniques. Furthermore, it competently performs exhaustive studies on complex, dynamic, nonlinear, and multilevel large-scale systems at both the macro- and micro-levels. The processes of SD model building and formation are as follows.

### 3.1.1. Model Framework

According to the data presented in Table 1, even though these 10 studies focus on different regions, objects, time periods, or models, most scholars agree that population size, economic growth, industrial structure, technological advancement, and energy consumption are the primary determinants of carbon emissions among all drivers. To summarize, the results obtained by previous studies demonstrate that carbon emissions are unambiguously linked to economic development, population size, and energy consumption. Based on the identified driving factors and their interactions, this paper developed four subsystems, which include the economic, demographic, energy, and carbon emissions subsystems.

**Table 1.** The summary of studies depicting the driving factors of carbon emissions.

| Scholars | Research Object | The Major Driving Factors of Carbon Emissions |
|---|---|---|
| Yang et al. [27] | Carbon emissions in Beijing | Population size, GDP, industrial structure, per capita income |
| Cui et al. [28] | Emissions from the construction industry in 30 provinces in China | Economic growth, industrial structure, urbanization, price volatility, marketization |
| Tan et al. [29] | Urban residential building sector in Gansu | Population size, GDP, industrial structure, energy efficiency, living space per capita |
| Lu et al. [30] | Carbon emissions from six major industrial sectors in China | GDP, energy structure, energy efficiency, floor space |
| Wu et al. [31] | Carbon emissions in the building sector | Development density, industry structure, energy structure, emissions factor |
| Li et al. [32] | Construction industry in Jiangsu Province | Output value, indirect carbon intensity effects, industry-scale effects |
| Zhang et al. [33] | Central and Eastern European countries | GDP, level of industrialization, measure of openness to the outside, technological innovation, energy structure |
| Chen et al. [34] | China's building sector | Demand effects, production structure effects, energy intensity effects |
| Du et al. [35] | China's building sector | Value creation effect, energy ratio, indirect carbon intensity, energy usage per unit value, output-scale effect |
| Li et al. [36] | Construction industry in Jiangsu Province | Indirect and operational CECIs, area factor, production value intensity factor, carbon emissions intensity |

### 3.1.2. Causal-Loop Diagram

Numerous frameworks for the driving factors of carbon emissions have been established in the literature to better investigate the peak value of carbon emissions produced by the building sector. The present paper incorporated all the factors that have an impact on carbon emissions, including population size, economic development, energy structure, and industrial structure, into the simulation system. All system factors were linked according to logical relationships. The four subsystems interacted and worked in concert to form a causal-loop diagram, which can be used to qualitatively analyze the causal relationship between the variables. The causal-loop diagram can be observed in Figure 3.

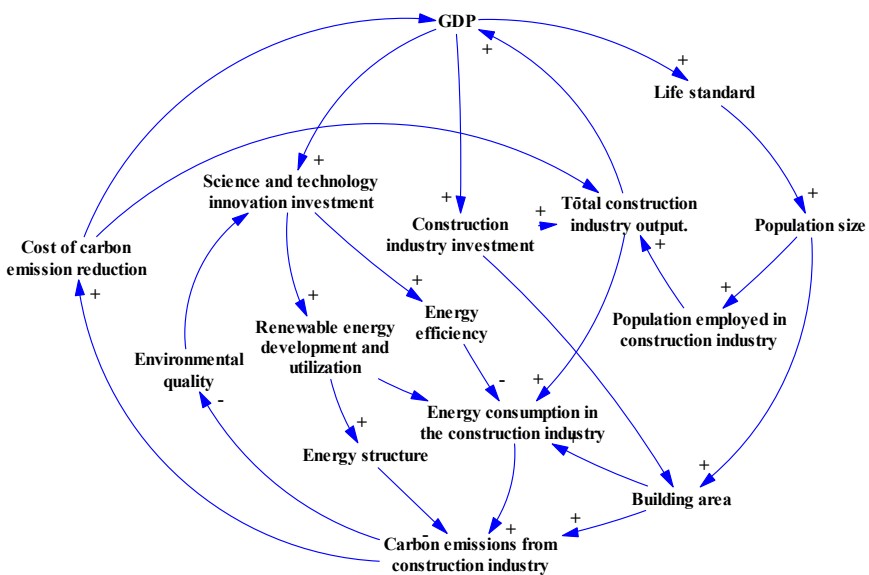

**Figure 3.** Causal-loop diagram of carbon emissions produced by the construction industry.

### 3.1.3. Stock Flow Diagram

To further explore the intricate influence mechanisms of carbon emissions produced by the construction industry, it is imperative to clarify the logical relationship between the system elements. Based on the causal relationship of the carbon emissions system,

this paper employed Vensim PLE64 software to draw the feedback stock flow diagram to quantitatively analyze the characteristic equations of the system model, which is depicted in Figure 4. Then, the initial parameters and functional equations of the model were established and input into the stock flow diagram. The system parameters and functional equations of the SD model were derived from surveys, statistics, and regression analyses. For instance, the functional relationship between energy efficiency and science and technology innovation investment was measured by linear regression using SPSS 26.0 software.

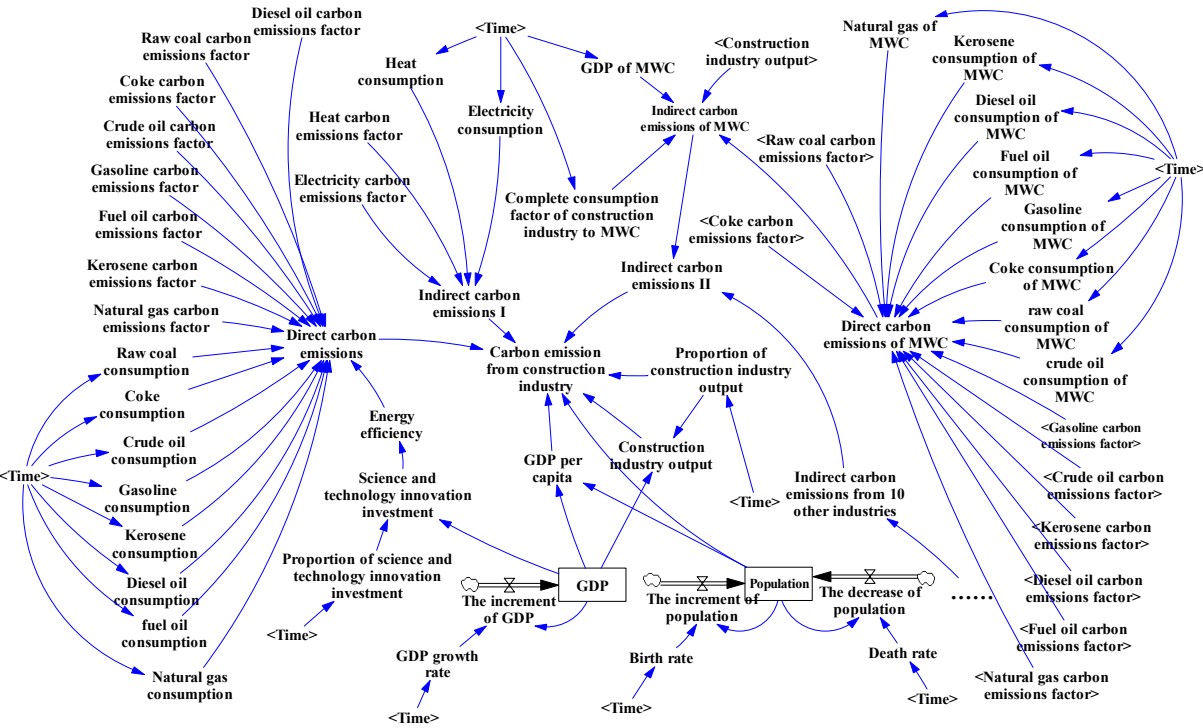

**Figure 4.** Stock flow diagram of carbon emissions produced by the construction industry.

### 3.2. Carbon Emissions Forecasting Model of the Construction Industry

Carbon emissions produced by the construction sector include both direct and indirect carbon emissions caused by the close connections with other related industries. Direct carbon emissions are produced by the industry itself; however, indirect carbon emissions are generated by other industries associated with the building industry. This paper divided the carbon emissions system of the construction sector into three parts working together without overlapping each other. The first part is direct carbon emissions, which are caused by the construction industry's direct energy consumption of raw coal, coke, crude oil, gasoline, kerosene, diesel oil, fuel oil, and natural gas. The second part is indirect carbon emissions, which are obtained from other industries that are driven by the construction sector. In order to improve the accuracy of the results, this paper categorized the carbon emissions produced by the electricity and heat in the construction sector as indirect carbon emissions I. Meanwhile, it computed the indirect carbon emissions II produced by 11 sectors related to the construction sector, which are: the mining and washing of coal (MWC); the extraction of petroleum and natural gas (EPNG); mining and processing of ferrous metal ores (MPFMOs); mining and processing of nonferrous metal ores (MPNMOs); petroleum refining, coking, and nuclear fuel processing (PRCNFP); manufacture of raw chemical materials and chemical products (MRCMCPs); manufacture of non-metallic mineral products (MNMPs); smelting and pressing of ferrous metals (SPFMs); smelting and pressing of nonferrous metals (SPNMs); manufacture of metal products (MMPs); and transport, storage, and postal services (TSPSs) [37]. In this study, we selected the MWC industry as the representative industry because of the diversity of related industries. The other ten

industries were sufficiently similar; therefore, they were not discussed in detail. Figure 4 depicts the MWC's forecasting model for carbon emissions. Direct carbon emissions and indirect carbon emissions I were calculated using the carbon emission factor method, while indirect carbon emissions II were accounted for using the input–output method.

### 3.2.1. Direct Carbon Emissions

Direct carbon emissions were calculated based on energy consumption values in the construction sector, using the formula in Equation (1):

$$C_{dir} = \Sigma E_i \times NCV_i \times A_i \times O_i \times \frac{44}{12} \tag{1}$$

where $C_{dir}$ is the direct carbon emissions (million tons (MT)). $E_i$ is the consumption of energy i in the building sector (MT of standard coal). $NCV_i$ is the average low calorific value of energy i. $A_i$ is the carbon content per unit calorific value of energy i. $O_i$ is the carbon oxidation rate of energy i.

### 3.2.2. Indirect Carbon Emissions I

Indirect carbon emissions I are obtained from electricity and heat consumption values in the building industry, which are calculated based on the carbon emissions formula:

$$C_{indirI} = Q_1 \times \beta_1 + Q_2 \times \beta_2 \tag{2}$$

where $C_{indirI}$ is the indirect carbon emissions released from electricity and heat driven by the construction industry (MT). $Q_1$ is the heat consumption rate of the construction industry (MT of standard coal). $\beta_1$ is the heat carbon emissions factor. $Q_2$ is the electricity consumption of the construction industry (billion kW·s). $\beta_2$ is the electricity emissions factor.

### 3.2.3. Indirect Carbon Emissions II

Due to the relationship evident between the construction industry and other industries, this paper selected carbon emissions generated by 11 industries associated with the construction industry as indirect carbon emissions II, which were estimated using the input–output method. The formula for calculating indirect carbon emissions II is as follows:

$$C_{indirII} = \sum_{j=1}^{11} \frac{C_j \times P \times y_j}{P_j} \tag{3}$$

where $C_j$ is the direct carbon emissions of the j industry; $P_j$ is the total output value of the j industry; P denotes the total output value of the construction industry; and $y_j$ denotes the complete consumption factor of the construction industry to the j industry.

The formula for calculating direct carbon emissions generated by 11 industries associated with the construction industry is as follows:

$$C_{11dir} = \Sigma G_i \times NCV_i \times A_i \times O_i \times \frac{44}{12} \tag{4}$$

$C_{11dir}$ are the direct carbon emissions from 11 industries, where i refers to the ith type of energy; $G_i$ is the consumption of the i type of energy by a related industry in the construction industry. Here, the meanings of $NCV_I$, $A_i$, and $O_i$ are the same as that presented in Formula (1).

The total carbon emissions produced by the construction sector are the sum of direct carbon emissions, indirect carbon emissions I, and indirect carbon emissions II. The formula for calculating total carbon emissions is as follows:

$$C = C_{dir} + C_{indirI} + C_{indirII} \tag{5}$$

*3.3. Model Testing*

Following the creation and measurement of the SD model, it was necessary to conduct a validity check to guarantee the credibility of the simulation results. As depicted in Figure 5, the results of the model validation can be represented by the relative error level. To verify its validity, this paper utilized the model to simulate and anticipate the relevant data from 2020 to 2030 under the baseline scenario using the historical data collected from 2000 to 2019. Subsequently, the SD model's effectiveness could be evaluated by contrasting the simulation values with the historical values obtained from 2000 to 2019. If the relative error was less than 10%, it indicated that the validity of the model was reliable [38]. Four indicators of carbon emissions released from the construction industry were selected to observe the magnitude of absolute errors between the historical and predicted values in 2000–2019, namely, direct carbon emissions, indirect carbon emissions I and II, and total carbon emissions. The results show that the relative errors of the simulation and historical values of the relevant variables remain within 7%, and all the average relative errors are maintained within 4%. There is a higher goodness of fit between the historical and simulation values. After being verified, the model performed well in its simulations and predictions.

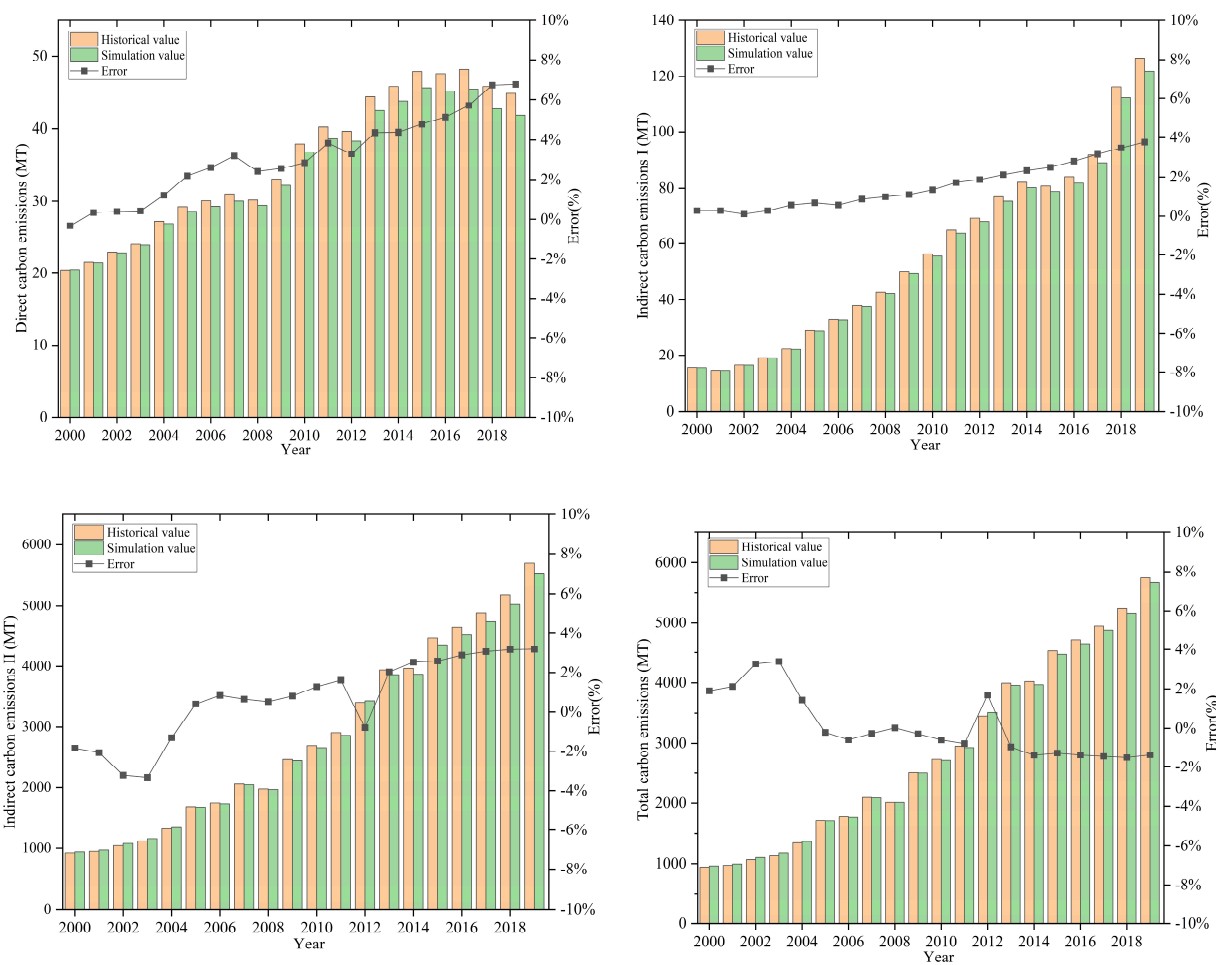

**Figure 5.** Error results for historical and simulation values.

*3.4. Scenario Setting*

To investigate the peaking status of the carbon emissions released by the construction sector from 2020 to 2030, this paper utilized the scenario analysis method based on the SD model to conduct simulations and make predictions. Firstly, the paper presented a baseline scenario to portray the characterization of carbon emissions in the absence of any

intervention. On this basis, we considered the following major impacting factors—economic growth, energy structure, industrial structure, and carbon emission policy factors— to create four single scenarios and one comprehensive scenario to simulate the dynamic changes in carbon emissions. The four single scenarios included economic growth, energy structure, industrial structure, and carbon emissions policy factor.

The economic growth scenario: In order to explore the effect of economic growth rate on carbon emissions released from the construction sector, we adjusted the parameters of the economic growth rate to predict carbon emissions, while maintaining other regulations. According to the literature concerning the driving factors presented in Table 1, economic growth is the leading contributor to carbon emissions in the building industry, which considerably boosts carbon emissions in China's building industry. Hence, the paper incorporated economic factors in the simulation analysis. Assume the characteristics of the economic growth rate scenario depicted in Table 2 are based on China's economic growth pattern.

**Table 2.** Economic growth rate scenario parameters in 2020–2030.

| Scenario | GDP Growth Rate |
| --- | --- |
| Low-speed growth rate | 5.3% (2020–2025) and 4.3% (2025–2030) |
| Medium-speed growth rate | 5.9% (2020–2025) and 5.1% (2025–2030) |
| High-speed growth rate | 6.5% (2020–2025) and 5.9% (2025–2030) |

The carbon emission policy factor scenario: The Chinese government has taken several steps to develop adequate carbon reduction technology. This study considered the effects of the improvements made to China's carbon reduction technology on carbon emissions produced by China's construction industry, where the policy factors characterized by the development of carbon emissions reduction technology were incorporated into the SD model. This was expressed using the percentage decrease in the carbon dioxide emissions factor of each energy source. As carbon emissions reduction technology advances, this paper set three types of carbon emissions policy factors at 2%, 5%, and 8% reductions from the base to observe how different emissions policy factors affect carbon emissions produced by the building industry, which are presented in Table 3.

**Table 3.** Carbon emissions policy factor scenario parameters in 2020–2030 ($kgCO_2/kg$ or $kgCO_2/m^3$).

| Type of Energy | Baseline | Lower than Baseline by 8% | Lower than Baseline by 5% | Lower than Baseline by 2% |
| --- | --- | --- | --- | --- |
| Raw coal | 1.9027 | 1.7505 | 1.8076 | 1.8646 |
| Coke | 2.8639 | 2.6348 | 2.7207 | 2.8066 |
| Crude oil | 3.024 | 2.7821 | 2.8728 | 2.9635 |
| Gasoline | 2.9287 | 2.6944 | 2.7823 | 2.8701 |
| Kerosene | 3.0372 | 2.7942 | 2.8853 | 2.9765 |
| Diesel fuel | 3.0998 | 2.8518 | 2.9448 | 3.0378 |
| Fuel oil | 3.1705 | 2.9169 | 3.0120 | 3.1071 |
| Natural gas | 2.1649 | 1.7505 | 1.8076 | 1.8646 |

The industrial structure scenario: The study explored the impact on carbon emissions produced by the construction industry by adjusting relevant industrial structure parameters, while maintaining other regulations, leaving them unchanged. According to the domestic industry development plan, the ratio of China's social investment in research and development to GDP will expand annually, reaching 2% by 2010 and more than 2.5% by 2020, with an average annual increase of 0.05%. On this basis, this paper set up three industrial structure scenarios based on various parameters, which are presented in Table 4.

**Table 4.** Industrial structure scenario parameters in 2020–2030.

| Industrial Structure Scenario | Proportion of Construction Industry Output | Proportion of Science and Technology Innovation Investment |
|---|---|---|
| Scenario I | 23% | 2% since 2010 and increases by 0.05% annually |
| Scenario II | 25% | 2.5% since 2010 and increases by 0.05% annually |
| Scenario III | 28% | 3% since 2010 and increases by 0.05% annually |

The energy structure scenario: To investigate the impact of energy structure adjustment policies on carbon emissions produced by China's building industry, the study determined energy structure scenarios that predicted carbon emissions while keeping other policies unchanged. Based on China's energy structure (which is "rich in coal, poor in oil, and poor in gas") and the premise of "stabilizing oil and increasing gas" in the 13th Five-Year Plan, the percentage of oil will follow the present trend, the percentage of coal will decrease, and the percentage of natural gas will increase. Three energy structure scenarios will be established, which are assumed to be high-, medium-, and low-speed scenarios relative to the base level, as presented in Table 5.

**Table 5.** Energy structure scenario parameters in 2020–2030.

| Scenario | Coal | Natural Gas |
|---|---|---|
| Low speed | Decrease by 10% in 2020–2030 | Increase by 10% in 2020–2030 |
| Medium speed | Decrease by 20% in 2020–2030 | Increase by 20% in 2020–2030 |
| High speed | Decrease by 30% in 2020–2030 | Increase by 30% in 2020–2030 |

The comprehensive scenario: Under the comprehensive scenario, the effects of all the individual policies accumulate. The comprehensive scenario is expected to include low energy consumption, medium energy consumption, and high energy consumption scenarios.

### 3.5. Data

The energy consumption of the construction industry and the energy consumption of industries related to the construction industry were obtained from the China Energy Statistical Yearbook. The total output value of the construction industry was obtained from the China Construction Statistical Yearbook, whereas the total output values of other related industries were obtained from the China Statistical Yearbook. The complete consumption factor of the construction industry for the other 11 industries was calculated according to the input–output table presented in the National Bureau of Statistics. It is updated once every five years; therefore, the data for the remaining years are estimations. The variables of population size, birth rate, death rate, GDP, and GDP growth rate were obtained from the China Statistical Yearbook. The carbon emissions factors for raw coal, coke, crude oil, gasoline, kerosene, diesel oil, fuel oil, and natural gas were calculated based on their carbon content per unit calorific value, carbon oxidation rate, and standard coal factor.

### 4. Results and Discussion

#### 4.1. Baseline Scenario Analysis

#### 4.1.1. Total Carbon Emissions

The results for the total carbon emissions produced by China's construction industry in 2000–2030 under the baseline scenario are presented in Figure 6. The total carbon emissions of the construction industry showed a stable increase on the whole; however, there was a decline as a result of the rigorous regulatory policy effected by the 2008 Olympic Games. Due to the economy's modest growth since 2014, there has been evidence of stable growth once again. As the new economic status continues to advance in 2021, the building sector will expand its reforms and accelerate the necessary transformations and upgrades. As a result, the increase in total carbon emissions will be regulated. Total carbon emissions

are predicted to reach 12,880.40 MT by 2030, which is almost 2.26 times the total amount of carbon emissions produced in the year 2019. With some regular adjustments, the annual growth rate of total carbon emissions produced by the construction industry has steadily declined.

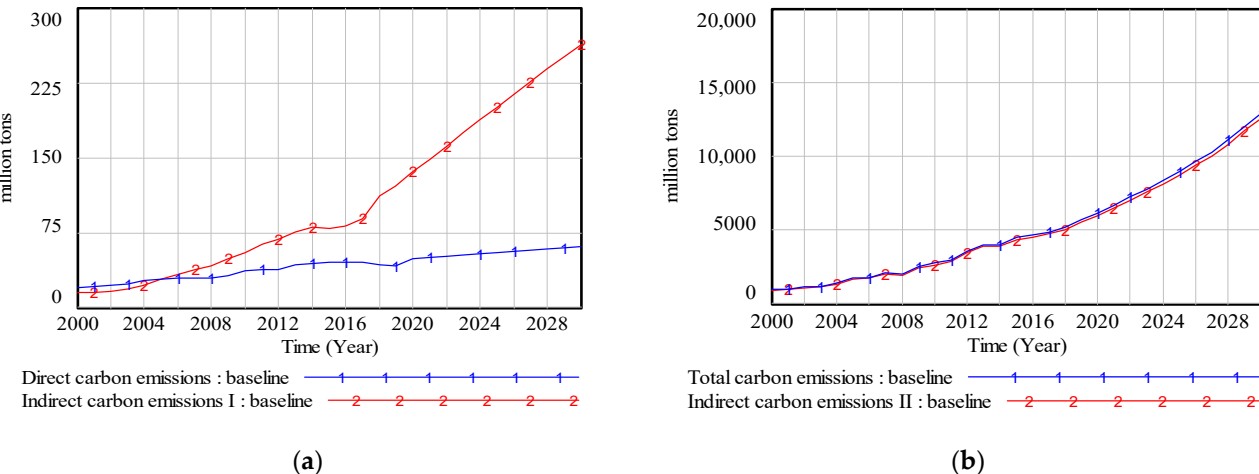

**Figure 6.** The carbon emissions prediction chart for each component. (**a**) Direct carbon emissions and indirect carbon emissions I. (**b**) Total carbon emissions and indirect carbon emissions II.

From the perspective of the components of carbon emissions, the results for each assessed segment appear to be increasing. The direct carbon emissions and indirect carbon emissions I are 69.06 and 263.27 MT, respectively, by 2030. Since 2005, indirect carbon emissions I have exceeded direct carbon emissions, and direct carbon emissions declined in 2018–2019 due to lower coal consumption. Even though direct and indirect carbon emissions I are increasing, their values remain far behind indirect carbon emissions II due to their lower initial emissions. Since 2000, the growth trend of indirect carbon emissions II has been consistent with that of total carbon emissions. The amount of indirect carbon emissions II is 12,555 MT, which accounts for over 96% of the total carbon emissions, and its proportion could reach 97.48% by the year 2030. It is an indisputable fact that the construction industry is a typical "apparently low-carbon, implicitly high-carbon" industry, with its carbon emissions primarily derived from related industries. As a result, the pivotal key to achieving an emissions reduction in the construction industry lies in indirect carbon emissions II, which has a significant carbon emissions reduction potential.

The results of the simulation indicate that the percentage of indirect carbon emissions produced by the associated industries will remain over 96%. Chen Y et al. (2016) also estimated in their study that the percentage of indirect carbon emissions would be approximately 97% in 2020, which matches our results [39]. Since direct carbon emissions only compose a small portion of the total emissions, it is evident that indirect carbon emissions produced by other related industries are the primary source and greatest contributor to the total carbon emissions. This is because the construction industry absorbs many of the products offered by other industries, which utilizes a considerable amount of energy and releases a lot more carbon dioxide into the air to create or process them, in turn resulting in a disproportionate share of the associated carbon emissions.

4.1.2. Indirect Carbon Emissions II

The carbon emissions forecasts for eleven industries are presented in Figure 7. It is evident that the construction industry displays the most remarkable pulling effect on the industries of PRCNFP, MRCMCP, SPFM, and SPNM. The four industries are estimated to add up to 2413.60, 1197.66, 3806.27, and 2617.04 MT by the year 2030, respectively. Despite this result, the growth rates of indirect carbon emissions produced by these four associated industries begin to slow down in 2018, with growth rates of 4.28%, 2.15%, 6.10%, and 7.91%

being achieved by 2030, respectively. Moreover, compared with the abovementioned four industries, the construction industry has a minor impact on the industries of MWC, MNMP, and TSPS, and the rest of the industries have less potential for mitigating indirect carbon emissions. As depicted in Figure 8, the growth rates of indirect carbon emissions generated by the eleven related industries are either positive or negative, and following a period of ups and downs, the growth rates have gradually settled down since 2020.

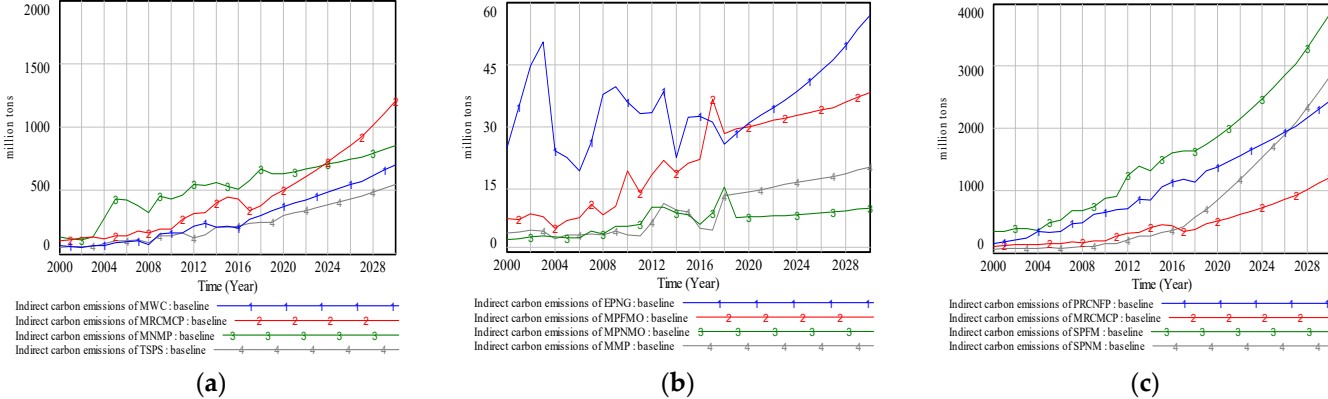

**Figure 7.** Carbon emissions forecasts for 11 industries. (**a**) Carbon emission of MWC, MRCMCP, MNMP, TSPS. (**b**) Carbon emission of EPNG, MPFMO, MPNMO, MMP. (**c**) Carbon emission of PRCNFP, MRCMCP, SPFM, SPNM.

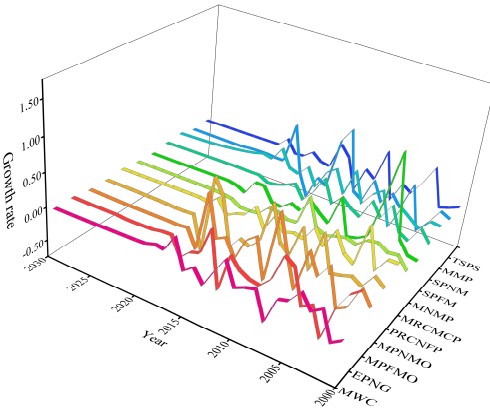

**Figure 8.** The growth rate of carbon emissions in 11 industries.

According to the proportion of carbon emissions for the eleven industries under examination (Figure 9), it is evident that the share of indirect carbon emissions of the aforementioned four industries is consistently higher than that of the other industries, which is due to their individual higher initial carbon emissions and faster growth rate, and their ratio is always higher than that of the other industries. The aforementioned four industries (except MRCMCP) are experiencing a gradual increase in their ratios. The share of the remaining seven industries is relatively stable, with little difference evident from the year 2019.

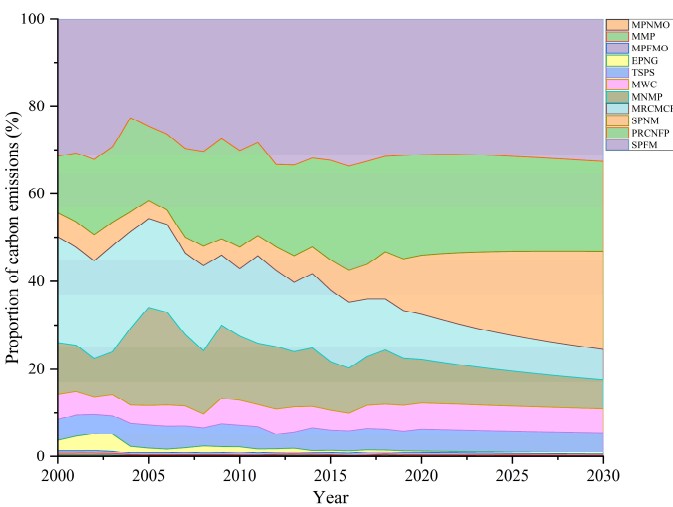

**Figure 9.** The proportion of carbon emissions for 11 industries.

Our results show that China's construction industry, especially its indirect carbon emissions, has enormous potential for carbon mitigation and is critical for China to meet its 2030 carbon peak commitment, which is consistent with other studies concerning the construction industry's being "apparently low-carbon and implicitly high-carbon" [40,41]. These studies showed that the greatest sources of indirect carbon emissions were the industries of PRCNFP, MRCMCP, SPFM, SPNM, and MWC. Our results and those of others support each other. On this basis, the government can set a more ambitious reduction goal to encourage low-carbon building developments in the building sector.

### 4.2. Single-Scenario Analysis

#### 4.2.1. Economic Growth Rate Scenario

In comparison to the baseline level, the growth rates for the economic growth scenarios are considered to be high, medium, and low. The simulation results obtained for the total carbon emissions in 2030 under the three scenarios are presented in Figure 10. The total carbon emissions will be reduced to 3.20–14.92% of the baseline scenario under the high-, medium-, and low-growth scenarios, respectively. It is clear that a higher growth rate will produce more carbon emissions. Additionally, indirect carbon emissions II are steadily on the decline, at 415.60–1925.10 MT compared to the baseline level. The results also show that the direct carbon emissions decrease slightly, while indirect carbon emissions I decrease by 17.18–19.70 MT.

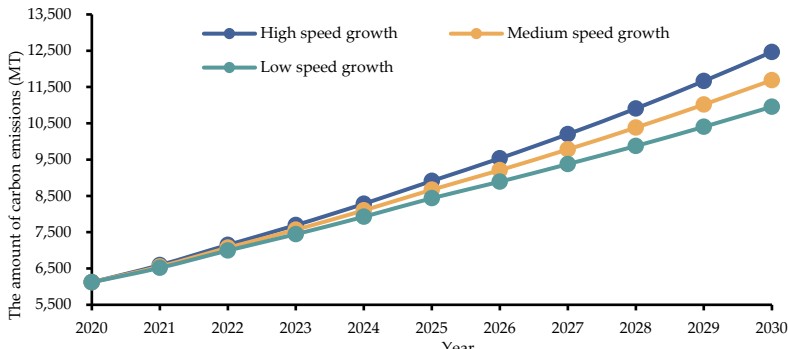

**Figure 10.** Result for carbon emissions under the economic growth scenario.

According to the abovementioned results, sustaining a high economic growth level is not necessarily favorable to the carbon emission peaks, which is in line with the simulation results obtained by Ma et al. (2019) and Li and Qin (2019) [42,43]. Since the establishment of China's economic reform and openness, carbon emissions have increased alongside the

country's thriving economy, despite the fact that the reduction effect is much lower than the driving effect.

### 4.2.2. Scenario of Carbon Emissions Policy Factor

The results for the three policy factor scenarios are illustrated in Figure 11. The scenario with the −8% policy factor has the lowest total carbon emissions in 2030, at 11,501.26 MT, which is a significant decrease of 1379.54 and 645.57 MT compared to the baseline and −2% policy factor scenarios, respectively. Both indirect carbon emissions II and direct carbon emissions are subjected to a continuous decline. Direct carbon emissions are reduced by 7.49–11.26 MT, whereas indirect carbon emissions II are reduced by 255.32–989.68 MT. Moreover, the reduction in indirect carbon emissions I is approximately 20 MT. Altering the parameters of the carbon emissions policy elements demonstrates that developing improved technology can help to reduce carbon emissions.

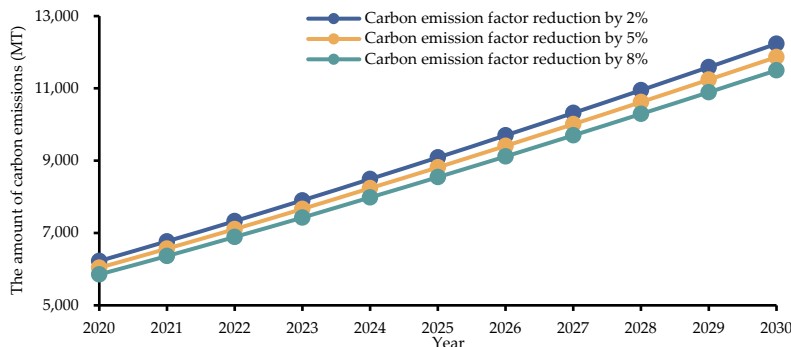

**Figure 11.** Result for carbon emissions under the carbon emissions policy factor scenario.

### 4.2.3. Industrial Structure Scenario

Figure 12 displays the outcomes of the carbon emissions simulation performed for the industrial structure scenarios. Under industrial structure scenarios I, II, and III, total carbon emissions in 2030 account for approximately 89.46–97.05% of the baseline, representing a minor decrease of 2.95–10.54%. In addition, direct carbon emissions and indirect carbon emissions II decreased to a certain level. Indirect carbon emissions II in 2030 compose approximately 77.84–80.17% of the baseline level under industrial structure scenarios I, II, and III, namely, a considerable decrease of 19.83–25.14%. Additionally, direct carbon emissions decreased by 9.31–13.29 MT, and indirect carbon emissions I decreased by 32.78–49.45 MT. It is evident from these results that adjusting the industrial structure is a slow process with a delayed impact on reducing carbon emissions.

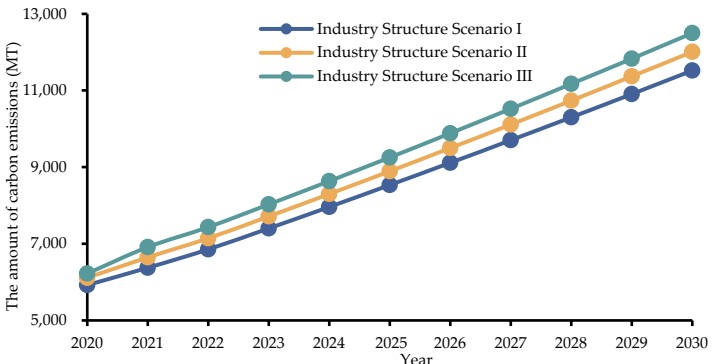

**Figure 12.** Result for carbon emissions under the industrial structure scenario.

### 4.2.4. Energy Structure Scenario

Figure 13 depicts the simulation results obtained for carbon emissions under the energy structure scenario. The total carbon emissions in 2030 account for approximately

59.44–68.57% of the baseline level under energy structure scenarios I, II, and III, namely, a striking decrease of 31.43–40.56%. The lowest total carbon emissions value is 7656.09 MT in 2030 under energy structure scenario III, which peaks in 2029, one year ahead of 2030. In particular, direct carbon emissions and indirect carbon emissions II significantly decreased. The direct carbon emissions in 2030 were reduced by 24.4–27.3% of the baseline level under scenarios I, II, and III, respectively. Under the three energy structure scenarios, indirect carbon emissions II in 2030 were reduced by 30.55–40.13% of the baseline level. It is apparent that optimizing the energy structure can considerably contribute to a significant reduction in the total carbon emissions produced by the building industry. The short-term effect of energy structure adjustments on the construction industry is obvious. The government and construction industry should make a joint effort to optimize the energy structure, encourage the use of natural gas instead of high coal consumption, and promote the construction industry's low-carbon development.

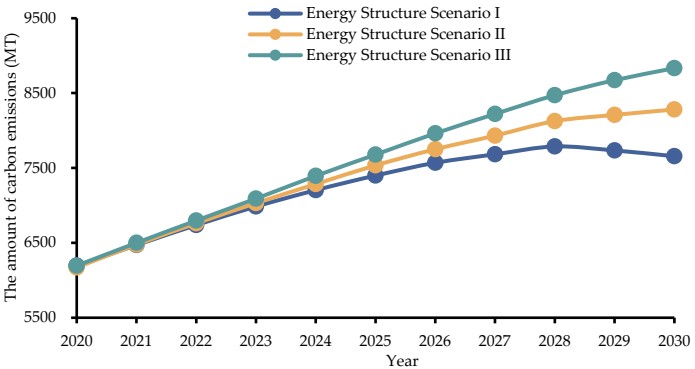

**Figure 13.** Result for carbon emissions under the energy structure scenario.

### 4.3. Comprehensive Scenario Analysis

Figure 14 displays the simulation results attained for carbon emissions in the comprehensive scenario. Compared to the adoption of single policies, the emissions reduction effect of the comprehensive scenario is superior to the single scenario. The total carbon emissions in 2030 merely account for nearly 43.27–73.73% of the baseline level under the medium- and high-energy consumption scenarios, namely, a striking decrease of 26.27–56.73%. Most importantly, under the low energy consumption scenario, the total carbon emissions were 5776.67 MT in 2028, while they were 5680.22 MT in 2029, a slight decrease compared to 2028, which demonstrates that the construction sector could attain a peak value in 2028 and meet the carbon peak target two years ahead of 2030. This result is mostly attributed to the simultaneous implementation of various carbon reduction initiatives, which considerably inhibit the increase in carbon emissions in China's building sector.

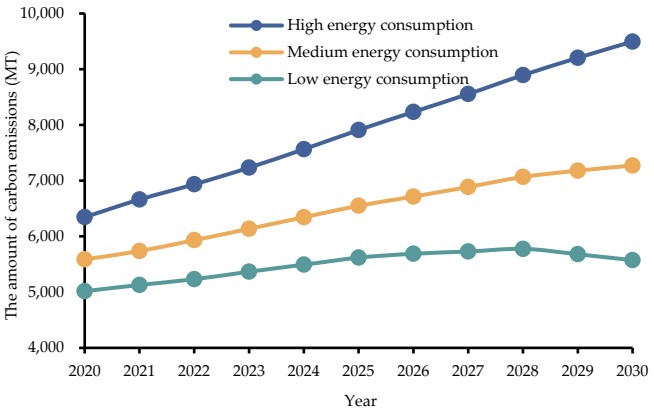

**Figure 14.** Result for carbon emissions under multiple scenarios.

The results obtained for the single- and multiple-policy scenarios are depicted in Figure 15. The carbon emissions reduction effect of the single-policy scenario varies with the diversity of initiatives. Their carbon emissions indicators are all better than the initial value. The emissions reduction effect of multiple-policy scenarios is the most obvious. As is indicated in the graph, the lowest curve is the comprehensive scenario curve, followed by the energy structure curve, carbon emissions policy factor curve, economic growth curve, and industrial structure curve. Driven by the implementation of multiple measures, the comprehensive scenario can lead to the greatest decline in total carbon emissions. This is better than four single-policy scenarios in terms of their ability to reduce carbon emissions.

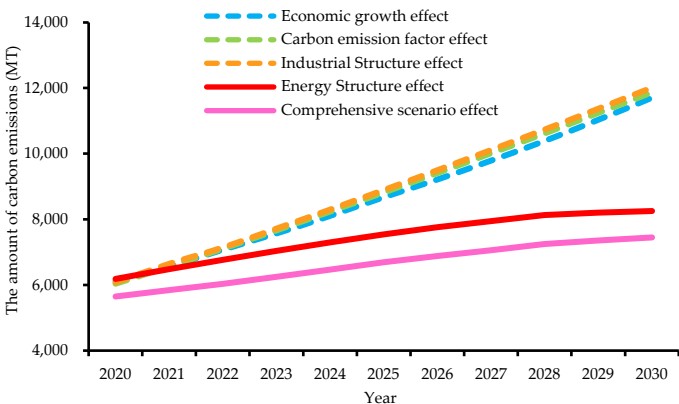

**Figure 15.** Result for carbon emissions reduction effect.

When compared to the emissions reduction effect of all single policies, the result reveals that energy structure > economic growth > carbon emission policy factor > industrial structure. Considering that only the comprehensive scenario outperforms the energy structure scenario, optimizing the energy structure is the most efficient path to low-carbon development. The improvement in economic structure is more beneficial to achieving low-carbon development. Regulating the policy factor is another straightforward method; however, the effect of industrial structure is comparatively moderate, because the influence of the coal production process takes a considerable amount of time to accumulate.

## 5. Conclusions

In this study, an SD model was established to investigate the prediction framework of China's construction industry, which considers direct carbon emissions (generated by the construction industry itself), indirect carbon emissions I (generated by electricity and heat), and indirect carbon emissions II (generated by 11 industries related to the construction industry). The dynamic change characteristic of carbon emissions was predicted under single- and multiple-scenario settings with altering economic growth rates, optimizing energy structure, adjusting industrial structure, and modifying carbon emission policy factors from 2020 to 2030. The study also investigated whether the building sector would be able to meet the carbon peak target by 2030. The main conclusions determined by our study are presented below.

(1) In the baseline scenario, the total carbon emissions produced by the construction industry will reach 12,880.40 MT by 2030, which is approximately 2.26 times the total amount of carbon emissions produced in 2019. The growth trend of indirect carbon emissions has nearly been consistent with that of total carbon emissions. Indirect carbon emissions produced by related industries comprise over 96% of the total carbon emissions present, whereas direct carbon emissions contribute only a little to the total. In particular, the construction industry has the most remarkable pulling effect on the industries of PRCNFP, MRCMCP, SPFM, and SPNM, which have the greatest potential to mitigate indirect carbon emissions. This shows that the construction industry is a typical "apparently low-carbon, but implicitly high-carbon" industry. Therefore, strengthening the control of

carbon emissions produced by related industries is essential to lowering carbon emissions. The government can establish a whole industry chain emissions reduction mechanism to promote emissions reductions by integrating resources and coordinating the actions of all industries and enterprises.

(2) Single scenarios have positive effects on carbon emissions reduction. However, only under energy structure scenario III can carbon emissions produced by the construction sector reach a peak before 2030. The lowest total carbon emissions will reach 7656.09 MT in 2030 under energy structure scenario III, which peaks in 2029, meeting the 2030 carbon peak target. This is the result of the effective improvement of the coal-based energy consumption structure, the control of coal consumption, and the substantial increase in natural gas consumption. By comparing the effects on the emissions reductions of single policies, the energy structure scenario has the most significant impact, while economic structure and policy factors still play a role in low-carbon development. However, the effect of the industrial structure is comparatively moderate, because the influence of the coal production process takes a considerable amount of time to accumulate. This implies that the Chinese government should prioritize improving the energy structure and promoting the application of clean and renewable energy and should encourage low-carbon technological upgrades and innovations, as well as optimize the industrial structure by encouraging enterprises to transform, upgrade, and develop low-carbon industries and production methods.

(3) The comprehensive scenarios produced the most positive effects on carbon emissions reduction. Additionally, we observed that only if four single policies are concurrently implemented will carbon emissions peak in 2028 at 5776.67 MT, two years ahead of schedule. This means that the comprehensive scenario effect is better than the four single-policy scenarios in terms of their ability to reduce carbon emissions. This is because it is the result of the implementation of multiple measures, instead of a single policy factor. Additionally, the adjustment of the energy structure is the most significant impact, followed by economic structure, policy factors, and industrial structure. This necessitates the joint efforts of governments, enterprises, and individuals to reduce carbon emissions through their cooperation and coordination.

There were certain limitations to this study, which can be improved in the future. Firstly, the paper predicted the carbon emissions produced by China's construction industry, not regional or provincial ones, which can be estimated for one day. Secondly, the availability of some of the data was limited. The complete consumption factor is updated once every five years; therefore, the data presented for the remaining years were estimated. In future research, the missing data can be estimated by using a certain computational procedure. Thirdly, this study established four different scenarios, and additional scenarios can be simulated in future research, such as carbon tax and trading scenarios.

**Author Contributions:** Conceptualization, X.W.; methodology, L.Q.; software, L.Q.; formal analysis, X.W. and L.Q.; writing—original draft preparation, X.W. and L.Q.; writing—review and editing, X.W., L.Q., H.X. and Y.W.; supervision, X.W. and L.Q. All authors have read and agreed to the published version of the manuscript.

**Funding:** The study was supported by the Shaanxi Social Science Foundation Project (2022D210) and the Social Science Prosperity Project of Xi'an University of Science and Technology (2022SZ01).

**Institutional Review Board Statement:** Not applicable.

**Informed Consent Statement:** Not applicable.

**Data Availability Statement:** Data sharing is not applicable to this article.

**Conflicts of Interest:** The authors declare that they have no known competing financial interests or personal relationships that could have appeared to influence the work reported in this paper.

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
