# Peer review of "Dynamic Scenario Predictions of Peak Carbon Emissions in China’s Construction Industry"

_sustainability, doi:10.3390/su15075922_

Round 1
Reviewer 1 Report
1) The introduction contextualizes the study's relevance well and presents a good theoretical reference regarding the existing studies to forecast carbon emissions in the construction industry and other sectors.
2) Replace the image in Figure 2 with a better-quality one.
3) Table 1 is very interesting because it brings the main factors that affect carbon emissions from existing studies in the literature.
4) Previous studies also employed the Vensim PLE64 software?
5) “In ord6er to improve accuracy, we categorize the carbon emissions produced by the electricity consumption and heat consumption in the construction sector as indirect carbon emissions” Check the sentence.
6) Replace Figure 5 with a better-quality one.
7) “If the relative error is less than 10%, it indicates that the validity of the model is reliable.” Insert reference that supports this claim.
8) “The finding shows that the relative errors of the simulation value and historical value of relevant variables are kept within 7%, and all the average relative errors are maintained within 4%.” Compare the error obtained in the study with those reported in the studies presented in the introduction.
9) Is the error not expected to be low considering that the data used contemplate this analysis interval?
10) “According to the literatures on the driving factors, economic development is the leading contributor to carbon emissions in the building industry, which greatly boosts carbon emissions in China's building industry” Mention literature in the text.
11) “As carbon emissions reduction technology advances, this paper sets three types of carbon emission policy factors at 2%, 5%, and 8% reduction from the base to see how different emission policy factors affect carbon emissions from the building industry”. Values ​​based on what?
12) The methodology needs to be better described and explained. For example, how were the scenarios for economic growth, carbon emission policy factor, industrial structure scenario, among others, defined?
13) Explain better what images a and b mean in Figure 6.
14) The conclusion is quite extensive. Synthesize sentences.
15) “There are certain restrictions on this study, which can be improved in the future. Firstly, the paper predicted the carbon emissions from China's construction industry rather than regional or provincial carbon emissions from China's construction industry. It can be estimated for one day. Secondly, the availability of some data is limited. The complete consumption factor is updated once every five years, thereby the data for the remaining years are estimated. In the future, missing data can be estimated by using a certain computation procedure. Thirdly, this study set four different scenarios, and more scenarios can be simulated in the future, such as carbon tax scenarios and carbon trading scenarios.” These limitations need to be presented in the methodology section of the article.
16) Insert in the conclusions the main contributions of the study.
Author Response
Dear reviewer 1,
Thank you for your comments and professional advice. These comments help to improve the academic rigor of our articles. Based on your suggestions and requests, we have made corrections to the revised manuscript. Also, the manuscript was reviewed and edited by the language service recommended by the journal, and we hope that our work will be improved again. In addition, we would like to clarify the following points.
Please see the attachment.

Reviewer 2 Report
The review comments are mentioned in the manuscript.

Author Response
Response to Reviewer 2 Comments
Dear reviewer 2,
We tried our best to improve the manuscript and made some changes to the manuscript. These changes will not influence the content and framework of the paper. And here we did not list the changes but marked in red in the revised paper. We appreciate for Editors/Reviewers’warm work earnestly and hope that the correction will meet with approval. We feel sorry for our poor writings, however, Thanks for your suggestion. We feel sorry for our poor writings, however, we use the polishing service agencies recommended by journals. And we hope the revised manuscript could be acceptable for you.
Point 1: grammer mistake.
Response 1: We feel sorry for our carelessness. The mistakes in grammar has been corrected in the following table.
|
Mistake |
Revision |
|
construction |
the construction |
|
reach |
the reach |
|
key |
a key |
|
variable |
variables |
|
Most |
the Most |
|
Exceed |
have exceeded |
|
Show |
are showen |
|
reduction |
the reduction |
|
illustrate |
are illustrate |
|
Or not |
/ |
|
play |
plays |
|
Or6der |
order |
|
whereas |
remove and |
|
Two |
in two |
Point 2: introduce STIRPAT, IPAT, SD, SDA.
Response 2: We feel sorry for our carelessness. I have given specific explanations for the abbreviations STIRPAT, IPAT, SD, and SDA.
Point 3:Merging the two figures in fig. 6.
Response 3: Thank you for this suggestion. There are four kinds of carbon emissions in the picture. Because the indirect carbon emissions are too large, the indirect carbon emissions I and the direct carbon emissions are too small, it is not effective to put them together. It seems more reasonable to separate a and b. Detailed explanations are given in the first and second paragraphs of 4.1.1Part.
Point 4:spaces.
Response 4: We feel sorry for our carelessness. I have corrected the spaces between references and word.
Point 5:million tons, billion tons.
Response 5: Thanks for your suggestion. I used abbreviations for million tons and billion tons except for the first time in the article.
Point 6:Improve Quality of the figture.
Response 6: We feel sorry for our carelessness. I have improved the quality of all my pictures.
Point 6:we
Response 6: We feel sorry for our carelessness. I have revised the first person we in the article.
Point 7:p
Response 7: We feel sorry for our carelessness. I have changed the p of the formula to P.
Point 8:Figure 5. The historical test results.
Response 8: We think this is an excellent suggestion. I have changed the “Figure 5. The historical test results” to“Figure 5. Error results between historical value and simulation value”.
Point 9:use one terminology either growth or envelopment.
Response 9: We think this is an excellent suggestion. I have changed all “development” to “growth”.
Point 10:getting repeated hence can be placed in the table headings.
Response 10: We think this is an excellent suggestion. I used headings for repeated contents in the table. I have modified Table 2-5.
Point 11:the units are not same in case of kg and m3 the factors will change or else the no
need of mentioning units.
Response 11: Thank you for your suggestion. Since natural gas is gas, its unit is kgCO2/m3. The unit of raw coal and coke is kgCO2/m3. So I think using kgCO2/kg or kgCO2/m3 is more reasonable
Point 12: How different from the medium speed as the value 30 % for coal and natural gas is same.
Response 12: I'm sorry I made a mistake about medium speed and high speed. I have revised them in Table 5.
Point 13:rewrite as emissions - I and replace everywhere, similarly for II.
Response 13: Thank you for this suggestion. I don't think it is reasonable to change the behavior of Emission I and Emission II, because I also use this form in scenario I, II, III. I don't think it is necessary to do this, and there are similar usages in other documents.
Point 14:Introduce the short forms used here as MWC, EPNG, MPFMO.
Response 14: Thank you for this suggestion. I have added a detailed explanation under the logo of Figure 2.
Point 15:All can be merged or at least a and c, why repeated.
Response 15: Thank you for this suggestion. The range of indirect carbon emissions of EPNG, MPFMO, MPNMO and MMP is 0-60 million tons. The range of indirect carbon emissions of PRCNFP, MRCMCP, SPFM, SPNM is 0-4000 million tons. If we merge them in a picture, it will look unreasonable and undistinct. Indirect carbon emissions of MRCMCP appears in Figures a and c for comparison purposes.
Point 16: Carbon emission (MT).
Response 16: Thank you for your suggestion. When I type millions tons into the equation unit, the graph looks like this. The reason is that these two diagrams were drawn by a system dynamics model, not by a drawing software. So I think it is meaningless to change the units of this graph. After reading a lot of reference materials on system dynamics, I found that the graphical unit of simulation is also millions of tons.
Point 17:rewrite.
Response 17: We think this is an excellent suggestion. I've changed the Abstract part of your mark to“Single and comprehensive scenarios have positive effects on reducing emissions; it was also observed that only under energy structure scenario III and comprehensive scenario III could carbon emissions released from the construction sector reach a peak value by 2030. The effects of emissions reductions as a result of single policies can be presented in the following order: energy structure, economic growth, carbon emissions policy factor, and industrial structure. All of the emissions reduction effects of multiple scenarios are superior to the single scenarios. ”
Point 17:appendix
Response 17: I am sorry that the appendix has been deleted due to space reasons, but I forgot to delete it in the original text.
Special thanks to you for your good comments.
Please see the attachment

Reviewer 3 Report
1. There are too many acronyms used by the paper starting from literature review session. Please provide full name of these acronyms, ex. STIRPAT model, IPAT model, CGE model, SD model?
2. Did the Table 1 take into consideration of other studies conducted outside China? Ex. Onat, N. C., Egilmez, G., & Tatari, O. (2014). Towards greening the US residential building stock: a system dynamics approach. Building and Environment, 78, 68-80. Proaño, L., Sarmiento, A. T., Figueredo, M., & Cobo, M. (2020). Techno-economic evaluation of indirect carbonation for CO2 emissions capture in cement industry: A system dynamics approach. Journal of Cleaner Production, 263, 121457.
3. Does this study take into consideration of the imported construction material’s carbon foodprint in the global market? Onat, N. C., & Kucukvar, M. (2020). Carbon footprint of construction industry: A global review and supply chain analysis. Renewable and Sustainable Energy Reviews, 124, 109783.
4. How does this paper’s method differ from the scenario simulation of carbon emission peak in China at Huo, T., Xu, L., Feng, W., Cai, W., & Liu, B. (2021). Dynamic scenario simulations of carbon emission peak in China's city-scale urban residential building sector through 2050. Energy Policy, 159, 112612? Du, Q., Shao, L., Zhou, J., Huang, N., Bao, T., & Hao, C. (2019). Dynamics and scenarios of carbon emissions in China’s construction industry. Sustainable Cities and Society, 48, 101556.
5. How did the authors generate Figure 3 and 4? Did the authors engage the local stakeholders in construction sector to determine these systematic dynamics? Or these were determined by scientists based on literature review? It is not entirely clear so far, please elaborate.
6. How can this paper’s scientific finding - Indirect carbon emissions from related industries make up over 96% of the total carbon emissions, whereas direct carbon emissions contribute little to the total carbon emissions, translate into policy making to reduce carbon emission in China from construction sector? For the reviewer, if the authors can answer this question, the contribution can be largely increased, instead of losing readers’ interest in these very complicated simulation models. In other words, the weakest part of this paper is in its conclusion. Please improve the concluding parts with communication or reference to other literature findings.
Author Response
Dear reviewer 3,
We feel great thanks for your professional review work on our article. As you are concerned, there are several problems that need to be addressed. According to your nice suggestions, we have made extensive corrections to our previous draft, the detailed corrections are listed below. We feel sorry for our poor writings, however, we use the polishing service agencies recommended by journals. And we hope the revised manuscript could be acceptable for you.
Please see the attachment

Round 2
Reviewer 1 Report
Accept
Reviewer 2 Report
The authors have addressed the comments, the manuscript can be accepted.